# Antiprotozoal Effects of *Pediococcus acidilactici*-Derived Postbiotic on *Blastocystis* Subtypes ST1/ST3

**DOI:** 10.3390/pathogens14070664

**Published:** 2025-07-05

**Authors:** Selahattin Aydemir, Yunus Emre Arvas, Mehmet Emin Aydemir, Fethi Barlık, Esra Gürbüz, Yener Yazgan, Abdurrahman Ekici

**Affiliations:** 1Department of Parasitology, Faculty of Medicine, Van Yüzüncü Yıl University, Van 65000, Türkiye; abdurrahman2400@gmail.com; 2Department of Molecular Biology and Genetics, Faculty of Science, Van Yüzüncü Yıl University, Van 65000, Türkiye; yunusemrearvas@yyu.edu.tr; 3Department of Veterinary Food Hygiene and Technology, Faculty of Veterinary Medicine, Harran University, Şanlıurfa 63200, Türkiye; aydemiremin23@harran.edu.tr; 4Van Health Services Vocational School, Van Yüzüncü Yıl University, Van 65000, Türkiye; fethibarlik@yyu.edu.tr; 5Department of Infectious Diseases and Clinical Microbiology, SBU Van Training and Research Hospital, Van 65000, Türkiye; dr.inanhazan@gmail.com; 6Department of Biophysics, Medical Faculty, Kastamonu University, Kastamonu 37750, Türkiye; yener8275@hotmail.com

**Keywords:** antiparasite, *Blastocystis*, organic acid, postbiotic, viability

## Abstract

*Blastocystis*, a common intestinal protozoan in humans, is associated with gastrointestinal disorders, irritable bowel syndrome, urticaria, and colorectal cancer. Its genetic diversity and potential for treatment resistance make it a focus of ongoing research. This study evaluated the in vitro antiprotozoal activity of a postbiotic derived from *Pediococcus acidilactici* as a natural alternative treatment. *P. acidilactici* cultures were grown in MRS broth under anaerobic conditions, and the postbiotic was collected and characterized for pH, yield, organic acid composition, and phenolic compound content. Human isolates of *Blastocystis* subtypes ST1 and ST3 were cultured in Jones’ medium and exposed to varying postbiotic concentrations for 72 h. Viability was assessed microscopically. The cytotoxic effect of the postbiotic-derived *P. acidilactici* was evaluated by investigating its impact on the viability of HT-29 cells using the Cell Counting Kit 8. The postbiotic showed a 7% yield and a pH of 4.52 ± 0.11. It contained seven different organic acids, predominantly lactic acid, and eleven phenolic compounds, with naringin as the most abundant. At 4.38 mg/mL, the postbiotic achieved over 94% inhibition and 100% inhibition at 8.75 mg/mL and above. A pH analysis confirmed that the inhibition was independent of the culture medium acidity. Cell viability was not affected at the postbiotic concentration showing 100% antiprotozoal activity (8.75 mg/mL). These findings suggest that the *P. acidilactici* postbiotic is effective on a mixed culture of ST1 and ST3 subtypes and holds promise as a safe, natural antiprotozoal agent. Further in vivo studies are needed to confirm this.

## 1. Introduction

*Blastocystis* is an anaerobic, unicellular, and enteric protozoan commonly found in humans. Its life cycle is not fully understood, and its transmission is known to occur via the fecal–oral route through cysts. Given the significant role of contaminated water in its transmission, the World Health Organization (WHO) has classified *Blastocystis* as a waterborne zoonoses [1]. Its prevalence in humans is estimated to be approximately 10% in developed countries and 50–60% in developing countries [2]. Molecular analysis of the small subunit ribosomal rRNA (SSU-rRNA) gene has identified 44 different subtypes (STs) in humans and animals [3]. Among these, ST1 to ST4 are the most prevalent in humans, accounting for approximately 90% of infections [4]. ST3 is the most frequently reported subtype, followed by ST1; however, the subtype distribution varies across populations and geographic regions [1].

Despite the numerous studies on *Blastocystis*, its pathogenicity remains one of the most debated issues. Its impact on human health is highly variable, with reports characterizing it as a pathogen, a commensal, and even a beneficial organism [5]. Although many studies have reported associations between *Blastocystis* and conditions such as gastrointestinal disorders, irritable bowel syndrome [6], urticaria [7], and colorectal cancer [2], its pathogenicity remains controversial. In addition to studies suggesting that its pathogenicity may be influenced by the host’s intestinal microbiota [8], evidence also indicates a potential association with specific *Blastocystis* subtypes [9].

Another area of controversy in *Blastocystis* infections is patient management. It is generally accepted that asymptomatic individuals infected with *Blastocystis* do not require medical treatment. However, emerging evidence supporting the pathogenic potential of *Blastocystis* underscores the need for treatment in certain cases [10]. For instance, in a reported case of a blastocystosis in a six-year-old child presenting with Stevens–Johnson syndrome, diarrhea, and abdominal pain, clinical symptoms, including skin lesions resolved following antiparasitic treatment [11]. Similarly, in a case report of *Blastocystis*-associated acute appendicular peritonitis in a nine-year-old child, it was noted that the patient’s C-reactive protein (CRP) levels did not improve without treatment for *Blastocystis* [12]. These findings underscore the importance of initiating *Blastocystis*-directed treatment in symptomatic cases where the parasite is identified as the sole infectious agent [13].

Although metronidazole is the first-line treatment for *Blastocystis* infections, studies have shown that it is not 100% effective, with some cases of blastocystosis persisting despite treatment [14]. On the other hand, alternative natural compounds for common parasitic infections, such as blastocystosis, have emerged as a promising treatment strategy in recent years, driven by the exorbitant production costs and potential adverse effects associated with conventional chemical therapies [10]. One such natural compound is postbiotics. Recent studies have introduced the concept of postbiotics by demonstrating that microorganisms can provide benefits even in their non-viable form [15]. Postbiotics are defined as secreted components and metabolites produced by probiotics that exert biological effects and mediate interactions between symbiotic bacteria and the host [16]. These metabolites contribute to the functional and therapeutic properties traditionally attributed to probiotics. Postbiotics contain a range of macro and micromolecules, including inactivated microbial cells; cell fractions; and metabolites such as organic acids, bacteriocins, and enzymes [15,17,18]. Some postbiotics have also been reported to exhibit antiparasitic effects [19,20].

The aim of this study was to evaluate the in vitro antiprotozoal activity of a postbiotic derived from *Pediococcus acidilactici* on *Blastocystis* and to investigate its potential as an alternative natural treatment agent.

## 2. Materials and Methods

This study was conducted between March and April 2025 at the Parasitology Laboratory of Van Yüzüncü Yıl University Faculty of Medicine. The study was approved by the Van Yüzüncü Yıl University Non-Interventional Clinical Research Ethics Committee (date: 28 February 2025; decision no: 2025/02-29).

### 2.1. Postbiotic Production Process

*P*. *acidilactici* (Lactoferm B-LC-78) culture was obtained from Chr. Hansen Laboratories (Copenhagen, Denmark). The strain was inoculated into De Man, Rogosa and Sharpe (MRS) broth (52.25 mg/mL) (Condalab, Madrid, Spain) and incubated under anaerobic conditions at 37 °C for 48 h. Following incubation, the culture was centrifuged at 4200× *g* for 10 min at 4 °C, and the supernatant was filtered using 0.45 µm filters (MF-Millipore, Merck KGaA, Darmstadt, Germany) [18]. The pH values were measured at 25 °C using a digital pH meter (EDT. GP 353) to assess the acidity of the postbiotic solution [21]. Subsequently, the freshly obtained postbiotic was lyophilized in a vacuum lyophilizer (Teknosem TRS-2-2, İstanbul, Türkiye) at −80 °C for 24 h. The total yield was calculated based on the difference between the initial and final weights of the sample [21].

### 2.2. Establishment of Experimental Groups

Three experimental groups were established as follows:Metronidazole group: Metronidazole (Nidazol, I.E. Ulagay Pharmaceutical Industry, İstanbul, Türkiye) was used as the reference drug. A stock solution was prepared by dissolving 0.8 mg of metronidazole in 1 mL of distilled water to achieve a final concentration of 0.05 mg/mL in the culture medium.Control group: Sterile distilled water was used as a negative control.MRS broth group: MRS broth was dissolved in distilled water to prepare a culture medium at a final concentration of 210 mg/mL. The MRS broth medium (210 mg/mL) without added bacteria was centrifuged at 4200× *g* for 10 min at 4 °C, and the supernatant was used by filtering using 0.45 µm filters. Subsequently, serial dilutions were performed with sterile distilled water to obtain final concentrations of 140, 70, 35, 17.5, and 8.75 mg/mL.*P. acidilactici* group: The fresh *P. acidilactici* postbiotic obtained after lyophilization was dissolved in sterile distilled water to prepare a stock solution at a concentration of 210 mg/mL. Subsequently, serial dilutions were performed with sterile distilled water to obtain final concentrations of 140, 70, 35, 17.5, and 8.75 mg/mL.

### 2.3. The Isolation of a Blastocystis Isolate

A *Blastocystis* isolate was obtained from a stool sample collected from a 58-year-old male patient who presented with diarrhea and abdominal pain at Van Yüzüncü Yıl University Dursun Odabaş Medical Center and tested positive for *Blastocystis*. A small portion of the stool (approximately the size of a grain of rice) was inoculated into Jones’ medium supplemented with fetal calf serum (FCS) (Capricorn, Ebsdorfergrund, Germany), 100 UI/mL of penicillin, and 100 μg/mL of streptomycin in 1.5 mL Eppendorf tubes supplemented and incubated at 37 °C [4,22]. After 72 h, vacuolar and granular forms of *Blastocystis* observed under a light microscope were washed with phosphate-buffered saline, and isolates were collected. *Blastocystis* cells were counted under a light microscope with a 40-times objective using a Thoma slide.

### 2.4. Antiprotozoal Activity of the Postbiotic on Blastocystis

Jones’ medium supplemented with FCS was dispensed into 1.5 mL Eppendorf tubes and inoculated with *Blastocystis* at a concentration of 5 × 10^4^ cells/mL. Then, 100 µL of the prepared solutions for each experimental group were added to the cultures, resulting in a final volume of 1600 µL per tube. Postbiotic, medium, and metronidazole concentrations were calculated based on their final concentrations following inoculation. After inoculation, culture tubes were incubated at 37 °C. Viable *Blastocystis* cells were counted at 24, 48, and 72 h after inoculation. Vital staining with 0.1% eosin and a hemocytometer (Thoma slide) along with a light microscope were used to evaluate antiprotozoal activity. For counting, the culture tubes were gently inverted, and 10 µL of the medium was mixed with 10 µL of the 0.1% eosin solution [22,23]. The preparations were examined under a light microscope using a Thoma slide with a 40-times objective, and the *Blastocystis* cells were counted. *Blastocystis* forms that stained orange were considered non-viable, while those that did not stain were considered viable (Figure 1). All processes were triplicated.

The percentage reduction of *Blastocystis* was calculated using the growth inhibition formula:(A − B/A) × 100,(1)
where A represents the mean number of viable cells in the control group, and B represents the mean number of viable cells in the treated tubes [24].

### 2.5. pH Measurements of the Culture Media

To assess whether *Blastocystis* inhibition was attributable to pH changes, the effect of the postbiotic on the pH of the culture medium was evaluated. pH values were measured for both the control group and the postbiotic-supplemented culture medium using a digital pH meter (EDT. GP 353), following the guidelines of the Association of Official Analytical Chemists (AOAC, 1990). Measurements were performed at room temperature by immersing the probe electrode directly into the culture medium. Before each measurement, a two-point calibration was performed using buffer solutions with pH values of 4.01 and 7.00.

### 2.6. Determination of Blastocystis Subtypes

After 72 h of incubation, 200 µL of a culture medium containing *Blastocystis* was collected for DNA extraction. DNA was isolated using a stool DNA isolation kit (Norgen Biotek Corp., Thorold, ON, Canada), following the manufacturer’s instructions. To identify *Blastocystis* subtypes ST1 to ST7, seven subtype-specific primer pairs targeting the SSU rRNA gene were used (Table 1) [25]. PCR reactions were prepared in a total volume of 25 µL, consisting of 5 µL of the Tag 5× Master Mix (containing 12.5 mM of MgCl2), 0.2 µM of each primer, and 2 µL of a template DNA. Amplification was performed using the Applied Biosystems SimpliAmp Thermal Cycler (Applied Biosystems, Foster City, CA, USA). The cycling conditions included 35 cycles of denaturation at 94 °C for 30 s, annealing at 57 °C for 30 s, and extension at 72 °C for 60 s. In addition to the PCR cycling protocol, an initial denaturation step at 94 °C for 4 min was performed prior to the first cycle, followed by a final extension at 72 °C for 10 min after the last cycle. To visualize the results, 15 µL of the amplified PCR products were subjected to gel electrophoresis and imaged using the UVP Gel documentation system.

### 2.7. Postbiotic Characterization

#### 2.7.1. Analysis of the Phenolic Compounds

Quantitative analysis of the phenolic compounds was performed using a high-performance liquid chromatography (HPLC) system equipped with a diode array detector (DAD) (Agilent Technologies, Santa Clara, CA, USA) [26]. The mobile phase consisted of two components: 83% phase A (purified water with 0.1% phosphoric acid) and 17% phase C (100% acetonitrile). The flow rate was maintained at 0.8 mL/min, with the column temperature set to 30 °C and an injection volume of 10 µL. Detection was performed using DAD, with analytes monitored at 300/200 nm and reference wavelengths set to 500/100 nm. The Phenolic Mix20 solution was used as the standard mixture, and quantitation was performed by comparing the peak areas of the identified phenolic compounds against calibration curves. The concentrations are expressed in mg/L based on the derived data.

#### 2.7.2. Analysis of the Organic Acid Content

The organic acid content of the postbiotic sample was determined using HPLC [27]. Separation was performed on a Hi-Plex H-type column (300 × 7.7 mm; PL1170-6830) at a constant temperature of 50 °C. The mobile phase consisted of 0.02 N sulfuric acid in an aqueous solution (100% water). Analyses were conducted under isocratic conditions with a flow rate of 0.6 mL/min and an injection volume of 10 µL. Detection was performed using a DAD set at a wavelength of 210 nm, with a reference wavelength of 400 nm. The resulting chromatographic data were manually integrated and evaluated. An organic acid Mix12 solution was used as the standard mixture, and quantification was performed by comparing the peak areas of the identified organic acids with their respective calibration curves.

### 2.8. Cell Viability Assay

To evaluate the cytotoxic effect of the postbiotic derived-*P. acidilactici*, its impact on the viability of HT-29 cells was investigated. Cell viability was measured using the Cell Counting Kit 8 (CCK-8) (Abbkine, Atlanta, Georgia, USA, Cat. No. KTA1020). HT-29 cells were seeded into 96-well plates at a density of 1 × 10^5^ cells/well. Absorbance at 450 nm was measured 72 h after treatment with postbiotics (0.55 mg/mL–52.5 mg/mL) using a microplate reader (Infinite PRO 200; Tecan Austria GmbH, Grodig, Austria). The experiment was repeated three times. Cell viability is expressed as a percentage relative to the control group (% control).

### 2.9. Statistical Analysis

All the experiments were conducted in two independent replicates, and the results are expressed as the mean ± standard error of the mean. Descriptive statistics were used to summarize the characterization data of the postbiotic. Analysis of variance (ANOVA) was used to evaluate differences in *Blastocystis* viability among the treatment groups. Multiple comparisons were performed using Tukey’s test, with statistical significance set at *p* < 0.05. The pH values of the culture media at different postbiotic concentrations were compared using the Kruskal–Wallis test. All statistical analyses were performed with IBM SPSS Statistics for Windows 21.0 (IBM Corp., Armonk, NY, USA).

## 3. Results

### 3.1. Postbiotic Production

The postbiotic was obtained with a yield of 7% and had a pH of 4.52 ± 0.11.

### 3.2. Antiprotozoal Activity of the Postbiotic on Blastocystis

The *P. acidilactici* postbiotic exhibited inhibitory effects on the in vitro growth of *Blastocystis*. The minimum concentration that inhibits 90% of organisms (MIC_90_) was determined to be 4.38 mg/mL, at which point inhibition of more than 94% was observed. Complete (100%) inhibition was achieved at concentrations of 8.75 and 13.13 mg/mL (Table 2).

Evaluation of the effects of the postbiotic concentration and incubation time on *Blastocystis* inhibition revealed statistically significant differences in the inhibition rates depending on both factors (*p* < 0.05). At a concentration of 2.19 mg/mL, the inhibition rate was 75.47% at 24 h, increasing to 84.06% at 48 and 87.21% at 48 and 72 h. Similarly, at 1.09 mg/mL, inhibition increased from 55.35% at 24 h to 83.85% at 72 h. The time-dependent increase in inhibition was statistically significant at both concentrations (*p* < 0.05) (Figure 2). These findings suggest that at lower concentrations, the antiprotozoal efficacy of the postbiotic increases over time, highlighting the importance of the incubation duration.

### 3.3. pH Measurement of the Culture Medium

The average pH of the culture pH medium in the control group was 7.43 ± 0.11. In the cultures containing postbiotic concentrations ranging from 0.55 to 4.38 mg/mL, the pH values remained above 7 (Table 3).

### 3.4. Blastocystis Subtype Identification

A PCR analysis revealed that the patient was co-infected with *Blastocystis* ST1 and ST3. Accordingly, both subtypes were simultaneously propagated in the culture medium (Figure 3).

### 3.5. Characterization of the Postbiotic

The analytical results indicate that the postbiotic was rich in organic acids, with seven distinct acids identified, of which lactic acid was the most abundant. The types and concentrations of organic acids present in the postbiotic are summarized in Table 4, and the corresponding chromatographic spectrum is presented in Figure 4.

In the phenolic compound analysis, eleven distinct phenolic compounds were identified in the postbiotic, with naringin, vanillin, chlorogenic acid, and caffeic acid being the most prominent. The compounds detected in the highest concentrations were naringin, vanillin, chlorogenic acid, *o*-coumaric acid, and caffeic acid, respectively. The types and concentrations of phenolic compounds present in the postbiotic are listed in Table 5, and the corresponding chromatographic spectrum is presented in Figure 5.

### 3.6. Cytotoxic Effect of the Postbiotic

No significant decrease in the viability rates of HT-29 cells was observed in postbiotic applications at concentrations of 0.55, 1.09, 2.19, 4.38, and 8.75 mg/mL. However, a 34.76% decrease in cell viability was observed at a concentration of 13.13 mg/mL (*p* = 0.001). The findings reveal that cell viability was not affected at the concentration showing 100% antiprotozoal activity (8.75 mg/mL) and that the LD_50_ value determined for HT-29 cells was well above the concentrations showing 100% antiprotozoal activity (Figure 6).

## 4. Discussion

The efficiency of postbiotic products is closely linked to the production process and the metabolic capacity of the microbial species used. Literature reports indicate that the efficiency of postbiotic production can vary depending on the microbial strain, medium composition, and fermentation conditions. However, yields typically fall within the range of 5–10% [21,28,29]. The 7% yield obtained in the present study is consistent with values reported in the literature. The acidic pH of the postbiotic indicates successful organic acid production and suggests high metabolic activity of the microbial culture [16]. The pH of the postbiotic produced in this study was 4.52 ± 0.11. Additionally, the *P. acidilactici* postbiotic was rich in organic acids and phenolic compounds. While the overall composition was comparable to previous studies [21,30], minor differences were observed. These variations are likely attributable to differences in the postbiotic production conditions, microbial strains, and analysis methods.

The surface layer protein SlpA, found in certain probiotics, plays a protective role in maintaining the gastrointestinal microbiota and the integrity of the intestinal mucosal barrier [31,32]. The most commonly used probiotics in clinical practice include bacterial strains from the genera *Lactobacillus*, *Bifidobacterium*, and *Streptococcus*, as well as the yeast *Saccharomyces*. One study reported that *Lactobacillus rhamnosus*, *L. lactis*, and *Enterococcus faecium* showed potential for use as prophylactic agents against *Blastocystis* colonization or as adjuvants in combination with standard drug therapies [33]. Another study found that *L. acidophilus* may be used as an adjuvant treatment alongside metronidazole in the management of blastocystosis [34].

Although the use of probiotics in the treatment of blastocystosis has been reported [33,34], their efficacy and safety, particularly in high-risk patients, remain controversial. As a result, postbiotics, which are bioactive metabolites produced by probiotics, have emerged as a promising alternative therapeutic approach [35]. Postbiotics offer several advantages over probiotics, including their safe use in the elderly and immunocompromised individuals without the risk of infection. Additionally, because they do not rely on viability, postbiotics retain their functionality even when administered alongside antibiotics [36]. In general, strains belonging to *Lactobacillus*, *Lactococcus*, *Enterococcus*, *Pediococcus*, *Staphylococcus*, *Leuconostoc*, and *Streptococcus* species are commonly used for postbiotic production [30].

*Pediococcus* spp., a member of the family *Lactobacillaceae*, and particularly *P. acidilactici*, are widely used in the food industry, animal husbandry, and medicine. Bacteriocins produced by *P. acidilactici* have been reported to inhibit the growth of pathogenic microorganisms in the host and function as signal-regulating peptides that modulate host health [37]. They have also been reported to play regulatory roles in maintaining intestinal flora balance [38].

Postbiotics have been reported to be effective in the treatment of protozoan infections, such as giardiasis, trypanosomiasis, and leishmaniasis [19]. Cuellar-Guevara et al. [20] demonstrated that postbiotics derived from *Lactobacillus* spp. inhibited the growth of *Entamoeba histolytica* protozoan under axenic conditions and induced morphometric changes in the cell membrane of the trophozoites. However, no studies to date have reported the effect of any bacterially derived postbiotic on *Blastocystis* viability. The current study is the first to demonstrate that postbiotics produced by *P. acidilactici* can inhibit the viability of *Blastocystis*. The *P. acidilactici* postbiotic exceeded the MIC_90_ threshold at a concentration of 4.38 mg/mL in the culture medium and achieved 100% inhibition of *Blastocystis* at 8.75 mg/mL. Consistent with previous reports [22], metronidazole at a concentration of 0.05 mg/mL also resulted in complete (100%) inhibition of *Blastocystis* in the present study. Although the inhibition activity of the postbiotic was observed at higher concentrations compared to metronidazole, which is currently used as the reference drug, the *P. acidilactici*-derived postbiotic appears promising as a natural therapeutic alternative, particularly given the well-documented side effects of metronidazole [39,40] and the emergence of metronidazole-resistant *Blastocystis* strains [41].

The growth of *Blastocystis* in Jones’ medium is strongly influenced by the pH of the culture environment [42]. Haziqah et al. [42] investigated the effect of pH on the viability of *Blastocystis* isolates from both humans and birds and reported that growth was suppressed at pH values below 3 for human isolates, below 4 for bird isolates, and below 5 in both types of isolates. Therefore, for in vitro studies, it is important to consider how the tested substance influences the pH of the culture medium when evaluating anti-*Blastocystis* activity. Substances that exert their inhibitory effect by lowering the pH of the culture medium may not be therapeutically effective in vivo, as they are unlikely to significantly alter the pH of the intestinal environment. In the present study, the pH value of the culture medium at the MIC_90_ concentration was above 7 and was similar to that of the control group. This suggests that the anti-*Blastocystis* effect of the *P. acidilactici* postbiotic is attributable to its bioactive compounds, independent of pH alteration.

Some studies have reported that the anti-*Blastocystis* activity of plant extracts varies depending on the *Blastocystis* subtype [4]. Accordingly, it is plausible that the effectiveness of postbiotics may also differ among *Blastocystis* subtypes. In the current study, the postbiotic derived from *P. acidilactici* was found to be effective against ST1, which is commonly associated with pathogenicity, and ST3, which is the most prevalent subtype in humans. Its observed effectiveness against a mixed culture of subtypes ST1 and ST3 supports the potential of this postbiotic as a therapeutic agent, although subtype-specific responses require further investigation.

The postbiotic produced in this study was found to be rich in organic acids, with seven distinct types identified. Organic acids are known to exert inhibitory effects on the growth of enteropathogenic microorganisms [43,44]. Indeed, organic acids have been widely reported to serve as acidifiers in animal feed, where they help modulate the intestinal microbiota and promote animal health through their antimicrobial activity [45]. Similarly, organic acids have been reported to exert inhibitory effects on *Blastocystis* infections [43,44]. However, no studies to date have examined the specific effects of the organic acids present in the postbiotic produced in this study on *Blastocystis* viability. On the other hand, oxalic [46], acetic, citric, lactic [47], and malic [48] acids have all been reported to exhibit antimicrobial activity. Propionic acid is a registered fungicide and bactericide commonly used in hay, grain storage, and drinking water for livestock and poultry [49]. Based on these findings, it is reasonable to suggest that the organic acids present in the postbiotic contribute to its anti-*Blastocystis* effect.

It should also be considered that the phenolic compounds present in the postbiotic may contribute to its anti-*Blastocystis* effect. Previous studies have reported that phenolic compounds suppress protozoan viability by disrupting cell membrane permeability [50]. Méabed et al. [51] reported that phenolic compounds in plant extracts exerted significant antiparasitic effects on *Blastocystis*, supporting the findings of the present study. In the present study, chlorogenic acid was identified as one of the predominant phenolic compounds in the postbiotic, and it has also been reported as a major component in certain plants known to exert anti-*Blastocystis* effects [51]. Moreover, naringenin, vanillin, *o*-coumaric acid, caffeic acid, and *p*-coumaric acid have also been reported to exhibit antiprotozoal activity. Specifically, both naringenin and *o*-coumaric acid have demonstrated leishmanicidal effects [52,53], vanillin has shown anti-*Toxoplasma* activity [54], caffeic acid has exhibited anti-malarial properties [55], and *p*-coumaric acid has exhibited anti-amoebic activity [56]. Based on these findings, it is plausible to suggest that the phenolic profile of the postbiotic contributes to its observed anti-*Blastocystis* activity.

Although postbiotics derived from *P. acidilactici* have been found to exhibit antiprotozoal activity, it is important to investigate their potential effects on the proliferation and viability of intestinal epithelial cells in order to evaluate their suitability for use as therapeutic agents. In one study, postbiotics derived from *P. acidilactici* were reported to increase the levels of short-chain fatty acids (SCFAs), such as acetic acid, propionic acid, and butyric acid in the intestinal tract, thereby supporting gut epithelial function, enhancing mucosal integrity, and maintaining intestinal homeostasis [57]. In this study, it was determined that the 8.75 mg/mL concentration, which was found to exhibit 100% antiprotozoal activity, did not have a negative effect on HT-29 cell viability. These findings suggest that *P. acidilactici* postbiotics may not only act against pathogens but also positively influence host gut health.

## 5. Conclusions

This study demonstrated that the postbiotic derived from *P. acidilactici* had strong inhibitory properties, achieving over 94% inhibition at a concentration of 4.38 mg/mL and complete inhibition at concentrations of 8.75 mg/mL and above. Importantly, these effects were independent of changes in the pH of the culture medium, suggesting that the antimicrobial activity was due to the bioactive components, such as organic acids and phenolic compounds, present in the postbiotic. Given the resistance issues and side effects associated with metronidazole, these findings highlight the potential of *P. acidilactici*-derived postbiotics as a promising natural alternative for the treatment of *Blastocystis* infections. However, in vivo studies are essential to validate these findings and explore the postbiotic’s mechanisms of action, bioavailability, and safety profile under physiological conditions.

## Figures and Tables

**Figure 1 pathogens-14-00664-f001:**
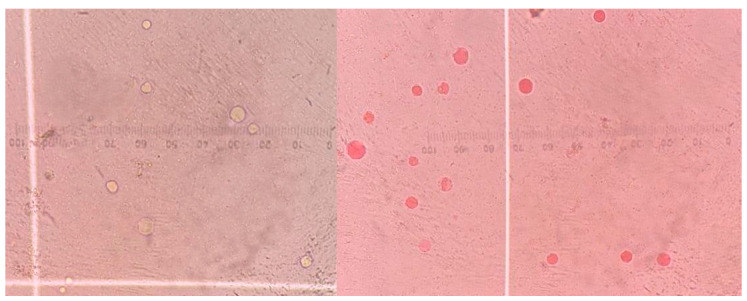
Thoma slide (400×) image showing viable (unstained) and non-viable (orange-stained) *Blastocystis* forms.

**Figure 2 pathogens-14-00664-f002:**
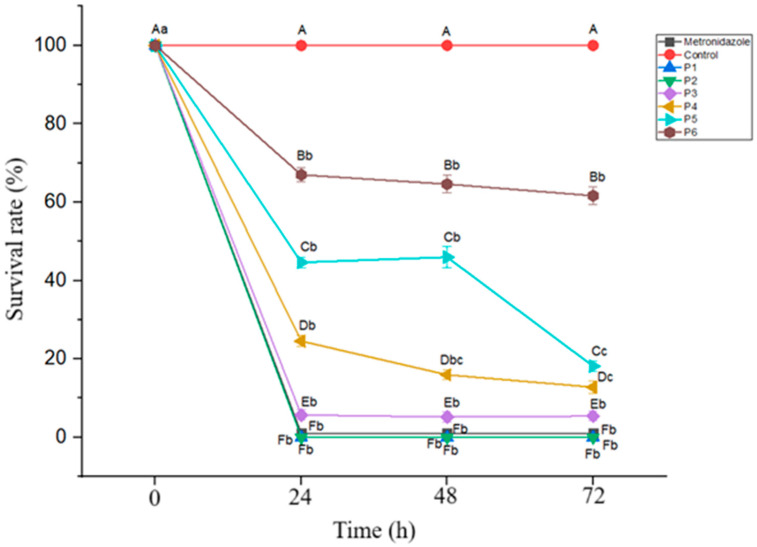
*Blastocystis* survival rates at different postbiotic concentrations over a 72 h incubation period. Mean values marked with different lowercase letters (a–c) indicate significant differences across the time points (*p* < 0.05), while different uppercase letters (A–F) indicate significant differences between the postbiotic concentrations (*p* < 0.05). (P: postbiotic, P1: 13.13 mg/mL, P2: 8.75 mg/mL, P3: 4.38 mg/mL, P4: 2.19 mg/mL, P5: 1.09 mg/mL, and P6: 0.55 mg/mL).

**Figure 3 pathogens-14-00664-f003:**
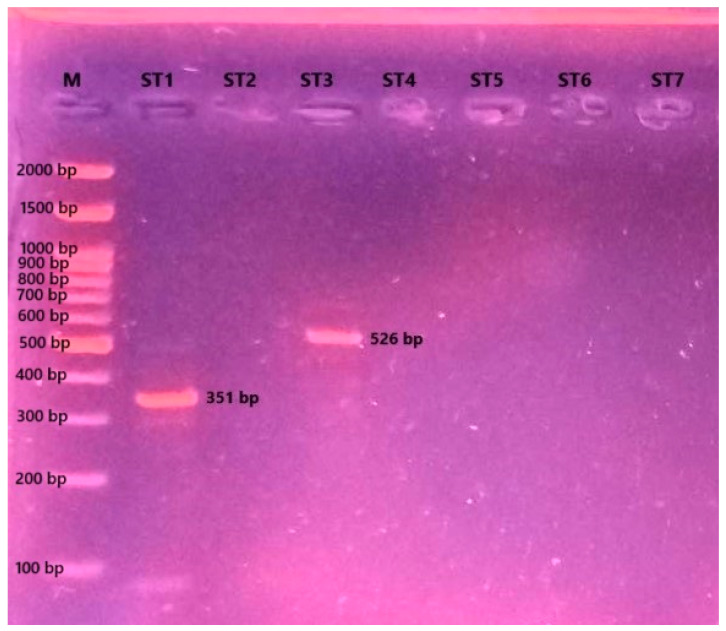
Agarose gel electrophoresis image showing the *Blastocystis* subtype-specific PCR products (M: molecular marker (Grisp Brand, Porto, Portugal); ST1-ST7: *Blastocystis* subtypes).

**Figure 4 pathogens-14-00664-f004:**
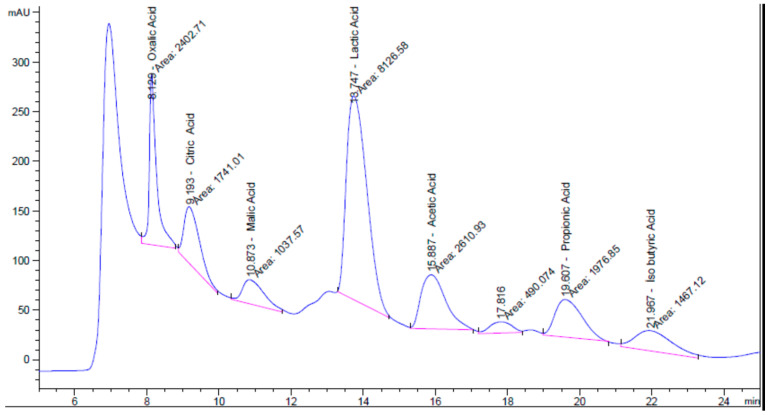
HPLC chromatogram showing the separation and detection of organic acids in the postbiotic sample.

**Figure 5 pathogens-14-00664-f005:**
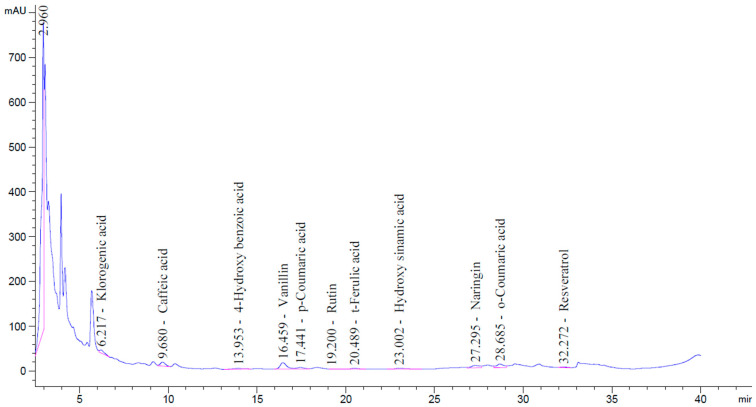
HPLC chromatogram showing the phenolic compounds identified in the postbiotic sample.

**Figure 6 pathogens-14-00664-f006:**
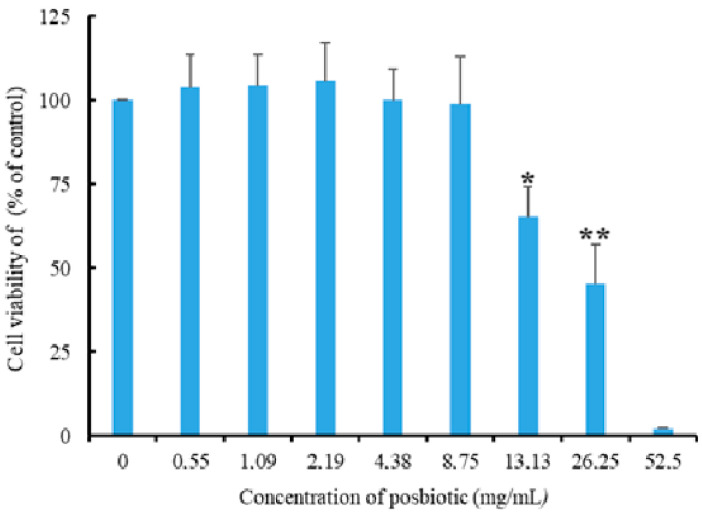
The effect of *Pediococcus acidilactici*-derived postbiotics on the viability of HT-29 cells (** and *: statistically significant difference compared to other groups; *p* = 0.001).

**Table 1 pathogens-14-00664-t001:** Primers used in the identification of *Blastocystis* subtypes ST1–ST7.

Subtype (ST)	Primers	Long (bp)
ST1	SB83F 5′-GAAGGACTCTCTGACGATGA-3′ SB83R 5′-GTCCAAATGAAAGGCAGC-3′	351
ST2	SB340F 5′-TGTTCTTGTGTCTTCTCAGCTC-3′ SB340R 5′-TTCTTTCACACTCCCGTCAT-3′	704
ST3	SB227F 5′-TAGGATTTGGTGTTTGGAGA-3′ SB227R 5′-TTAGAAGTGAAGGAGATGGAAG-3′	526
ST4	SB337F 5′-GTCTTTCCCTGTCTATTCTGCA-3′ SB337R 5′-AATTCGGTCTGCTTCTTCTG-3′	487
ST5	SB336F 5′-GTGGGTAGAGGAAGGAAAACA-3′ SB336R 5′-AGAACAAGTCGATGAAGTGAGAT-3′	317
ST6	SB332F 5′-GCATCCAGACTACTATCAACATT-3′ SB332R 5′-CCATTTTCAGACAACCACTTA-3′	338
ST7	SB155F 5′-ATCAGCCTACAATCTCCTC-3′ SB155R 5′-ATCGCCACTTCTCCAAT-3′	650

**Table 2 pathogens-14-00664-t002:** Percentage (%) inhibition of *Blastocystis* by the *P. acidilactici* postbiotic over time and at varying concentrations.

Group	Concentration (mg/mL) *	After 24 h		After 48 h		After 72 h		
		Number ** ± SD	%Inhibition	Number ** ± SD	%Inhibition	Number ** ± SD	%Inhibition	*p* ***
Control	0.00	13.25 ± 4.99	0.00	40.25 ± 13.89	0.00	64.50 ± 20.40	0.00 ^a^	0.001
Metronidazole	0.05	0.00 ± 0.00	100.00 ^a^	0.00 ± 0.00	100.00	0.00 ± 0.00	100.00 ^g^	
MRS broth medium	0.55	13.67 ± 3.79	0.00	42.33 ± 3.06	0.00	64.50 ± 6.03	0.00 ^a^	
	1.09	13.00 ± 4.58	1.88	41.30 ± 4.93	0.00	67.00 ± 8.89	0.00 ^a^	
	2.19	14.00 ± 2.00	0.00	40.67± 3.06	0.00	68.33 ± 8.50	0.00 ^a^	
	4.38	12.67 ± 4.04	4.38	38.33 ± 3.51	4.77	61.67 ± 11.93	4.38 ^a^	
	8.75	12.33 ± 3.21	6.94	36.67 ± 3.79	8.89	60.33 ± 8.02	6.46 ^a^	
	13.13	11.67 ± 2.08	11.92	35.67 ± 3,06	11.37	55.00 ± 6.24	14.73 ^b^	
Postbiotic	0.55	8.88 ± 2.65	33.02	26.00 ± 1.75	35.40	39.80 ±10.80	38.37 ^c^	
	1.09	5.92 ± 2.25	55.35	18.50 ± 4.75	54.04	10.40 ± 2.20	83.85 ^d^	
	2.19	3.25 ± 0.75	75.47	6.42 ± 3.59	84.06	8.3 ± 1.1	87.21 ^e^	
	4.38	0.75 ± 0.25	94.34	2.08 ± 0.58	94.82	3.5 ± 0.5	94.57 ^f^	
	8.75	0.00 ± 0.00	100.00 ^a^	0.00 ± 0.00	100.00	0.00 ± 0.00	100.00 ^g^	
	13.13	0.00 ± 0.00	100.00 ^a^	0.00 ± 0.00	100.00	0.00 ± 0.00	100.00 ^g^	

* Postbiotic and metronidazole concentrations were calculated based on their final concentrations after culture inoculation; ** ×10^5^ viable *Blastocystis*/mL; *** according to inhibition values after 72; ^a–g^: shows differences between concentration groups.

**Table 3 pathogens-14-00664-t003:** pH values of the culture medium at different postbiotic concentrations.

Concentration (mg/mL)	pH Value	*p* *
0 (Control)	7.43 ± 0.11 ^a^	0.001
0.55	7.23 ± 0.07 ^a^	
1.09	7.28 ± 0.05 ^a^	
2.19	7.23 ± 0.08 ^a^	
4.38	7.09 ± 0.03 ^a^	
8.75	6.26 ± 0.1 ^b^	
13.13	5.71 ± 0.1 ^c^	

* Significance levels according to Kruskal–Wallis test; ^a^, ^b^, ^c^: shows differences between concentration groups.

**Table 4 pathogens-14-00664-t004:** Composition and concentrations of the organic acids identified in the postbiotic sample.

Organic Acid	Retention Time (min)	Peak Area (mAU·s)	Concentration (ng/µL)
Oxalic Acid	8.129	2402.71	188.95
Citric Acid	9.193	1741.01	2102.46
Malic Acid	10.873	1037.57	1414.62
Lactic Acid	13.747	8126.58	14,040.20
Acetic Acid	15.887	2610.93	10,050.10
Propionic Acid	19.607	1976.85	529.55
Isobutyric Acid	21.967	1467.12	321.87

**Table 5 pathogens-14-00664-t005:** Composition and concentrations of phenolic compounds identified in the postbiotic sample.

Compound Name	Retention Time (min)	Peak Area (mAU·s)	Concentration (ng/µL)
Chlorogenic Acid	6.217	137.621	8.515
Caffeic Acid	9.680	177.898	5.024
4-Hydroxy benzoic Acid	13.953	76.895	2.561
Vanillin	16.459	370.830	12.85
*p*-Coumaric Acid	17.441	116.670	3.095
Rutin	19.200	10.501	0.298
t-Ferulic Acid	20.489	52.024	1.577
Hydroxycinnamic Acid	23.002	72.441	1.686
Naringin	27.295	194.189	19.465
*o*-Coumaric Acid	28.685	212.727	5.030
Resveratrol	32.272	26.048	0.666

## Data Availability

The original contributions presented in this study are included in the article. Further inquiries can be directed to the corresponding author.

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
