# Peer review of "Antiprotozoal Effects of Pediococcus acidilactici-Derived Postbiotic on Blastocystis Subtypes ST1/ST3"

_pathogens, 2025, doi:10.3390/pathogens14070664_

Round 1
Reviewer 1 Report
Comments and Suggestions for Authors
Dear authors, the manuscript titled " Antiprotozoal effects of Pediococcus acidilactici-derived post-2 biotic on Blastocystis subtypes ST1 and ST3", must be revised and restructured, with the aim that the results obtained have the desired impact on possible readers and that are reliable, universal and reproducible, so I request you to respond to the following observations:
- Check that “in vitro” and “in vivo” are written correctly throughout the document
- ¿How are postbiotic concentrations obtained?
- ¿How many repetitions are made for each treatment?
- In the Antiprotozoal Activity of the Postbiotic on Blastocystis assay, a volume of 1600 microliters were reached in a container that has a capacity for 1500 microliters
- ¿What is the name of the method or technique used to evaluate Antiprotozoal Activity?
6.¿ What technique was used to count 50000 Blastocystis
- Enter the molecular weight of the markers
- Only two bibliographic citations were identified in materials and methods, ¿the rest of the methodology is new and proposed by the authors?
- ¿How was MIC90 determined if only Blastocystis viability was determined?
- ¿Why were the results of tables 4 and 5 not analyzed statistically?
- The information in figure 5 is confusing, look for a better way to present the results
- Discuss the results in the order in which they are presented or change the order of the methodology and results so that they coincide with the discussion
- Adequately discuss the results of identified phenolic compounds
- The conclusion must be written considering the aim and the significance of the results obtained
I could not comment on this matter.
Reviewer 2 Report
Comments and Suggestions for Authors
The article submitted for peer review needs to be much improved to be published in Pathogens MDPI.
This study evaluated the in vitro antiprotozoal activity of a postbiotic derived from Pediococcus acidilactici as a natural alternative treatment, in which contained seven different organic acids, predominantly lactic acid, and eleven phenolic compounds, with naringin as the most abundant. Authors suggested that P. acidilactici postbiotic is effective against ST1 and ST3. However, there are opportunities to enhance the clarity and precision of the writing throughout the manuscript.
Major
- The results showed that the P. acidilactici postbiotic worked against Blastocystis ST1 and ST3. The authors also suggested that postbiotics don't need to be alive to work, meaning they remain effective even when used in conjunction with antibiotics. However, the current data only shows effects in the lab. Earlier studies using mice sometimes found that postbiotics didn't work as well as giving live bacteria. To fully support the title, "Antiprotozoal effects of Pediococcus acidilactici-derived postbiotic on Blastocystis subtypes ST1 and ST3," we strongly recommend adding animal experiments. This will help connect the lab findings to what might happen in a living body.
- While the authors appropriately highlight the widespread application of Pediococcus species, particularly P. acidilactici, across the food, animal husbandry, and medical sectors, it is important to evaluate the potential impact of P. acidilactici postbiotic on the proliferation and viability of gut epithelial cells. This additional assessment would provide crucial insights into the safety profile and potential beneficial or detrimental effects of the postbiotic on the host intestinal environment.
- The methodology for counting viable Blastocystis cells, specifically the use of 0.1% eosin solution at 24, 48, and 72 h post-inoculation, requires clear citation of an associated reference. Historically, the trypan blue dye exclusion method has been the standard for assessing protozoal viability. If the authors have developed a novel counting methodology, it is incumbent upon them to provide comprehensive validation data demonstrating its accuracy, reliability, and comparability to established techniques.
- Section 2.4, "Isolation of the Blastocystis Isolate," lacks a comprehensive description of the methodology employed for isolating Blastocystis from the patient's stool sample. The current description states the inoculation of stool into culture medium without mentioning the use of antibiotics to inhibit bacterial overgrowth. It is widely acknowledged that antibiotic treatment is a critical step for the axenic isolation of Blastocystis species, a practice established by the seminal work of Zierdt and Williams. Furthermore, reference (21) indicates that the authors have previously isolated Blastocystis from other patients and conducted drug activity evaluations in 2024. Therefore, the manuscript must clearly delineate the specific isolation methods utilized in this study and provide a compelling rationale for employing a new isolate from the 58-year-old patient, rather than utilizing the previously characterized isolate from 2024. The consistency of using the same isolate across different experiments is paramount for ensuring comparability and reliability of drug activity assessments.
- While PCR analysis successfully identified a co-infection of Blastocystis ST1 and ST3 in the patient sample, the authors did not subsequently isolate these subtypes individually from the mixed culture. This presents a critical limitation, as ST1 and ST3 are known to exhibit distinct cytotoxic effects on host cells and animals, and may also possess differential sensitivities to postbiotic treatments. To align with the study's title and provide a more precise understanding of the postbiotic's inhibitory effects, it is important to clarify whether the observed inhibition targets ST1, ST3, or both. We strongly recommend that the authors undertake the isolation of ST1 and ST3 from their samples and evaluate the inhibitory effect of the postbiotic on each subtype separately.
- According to the Centers for Disease Control and Prevention (CDC), among the nine Blastocystis STs identified in humans to date, ST1, ST2, ST3, and ST4 are the four most prevalent. Given that the authors have successfully isolated Blastocystis from other patients in previous studies (as evidenced by their references), it would significantly enhance the comprehensiveness and translational relevance of this research to additionally evaluate the effect of the postbiotic on ST2 and ST4. This expanded assessment would provide a more robust understanding of the postbiotic's inhibitory spectrum against the most prevalent human-associated Blastocystis subtypes.
- The culture of Blastocystis usually needs an environment with no oxygen. The authors put stool samples into 1.5 mL Eppendorf tubes, but they didn't explain how they transferred or continued the cultures (subculture). Do closed 1.5 mL Eppendorf tubes create a good oxygen-free environment?
Minor
“The postbiotic was obtained with a yield of 7% (7/1 mL) and had a pH of 4.52 ± 0.11” Please clarify “(7/1 mL).”
Round 2
Reviewer 1 Report
Comments and Suggestions for Authors
Dear authors, please respond to the following observations:
1.- Check that the complete document P. acidilactici is written correctly
2.- In lines 280 and 281 the letter format is different
Author Response
Thank you for approving our article. The grammatical errors you mentioned have been corrected.
Reviewer 2 Report
Comments and Suggestions for Authors
Major
1.
We thank the authors for their transparent explanation regarding the challenges with long-term culturing.
However, the explanation raises a more significant concern regarding the reproducibility and generalizability of the findings. The authors state that previous isolates were lost due to "technical limitations in long-term subculturing under the available laboratory conditions." A review of the authors' publication history reveals a recurring pattern of isolating a new clinical strain for each new study that evaluates therapeutic activity.
This recurring necessity to use new, single-patient isolates suggests a systemic issue with the stability and robustness of the culture methodology itself. Consequently, the reported antiprotozoal effects may be specific only to the unique and transient isolate from the 58-year-old patient. This approach severely limits the ability of the international research community to independently reproduce, verify, or build upon the reported results.
To address this critical limitation and ensure the study's findings are both robust and meaningful to the wider scientific community, we strongly insist that the authors validate their results using a standardized, globally accessible reference strain. Utilizing a strain from a recognized culture collection, such as the American Type Culture Collection (ATCC), is standard practice for ensuring experimental repeatability.
Therefore, we recommend that the activity of the postbiotic be evaluated on a well-characterized Blastocystis reference strain (e.g., an isolate available from ATCC). This will provide the necessary evidence that the observed effects are not isolate-specific and will establish a reliable foundation for future research in other laboratories. Without such validation, the conclusions drawn from a single, non-maintainable isolate remain preliminary, and their broader scientific impact is questionable.
2.
The authors report achieving complete (100%) inhibition of Blastocystis viability at a concentration of 8.75 mg/mL. However, this conclusion warrants careful re-evaluation due to significant methodological limitations. The authors confirmed the presence of both Blastocystis ST1 and ST3 subtypes using PCR, yet they were unable to isolate these subtypes to create single-subtype cultures. Furthermore, as acknowledged in previous correspondence, the isolated culture system is not stable for long-term subculturing. These two points lead to a critical flaw in interpretation:
The initial ratio of ST1 to ST3 in the culture is unknown.
It is difficult to determine if the postbiotic treatment affected both subtypes equally, or if it selectively inhibited one subtype over the other. The observed 100% inhibition could mask a differential susceptibility between the subtypes.
Therefore, the study does not demonstrate the effect on ST1 and ST3 populations, but rather the effect on a single, uncharacterized mixed population of Blastocystis. The current claims appear to overstate the results. The only conclusion that can be robustly drawn from the data is that the postbiotic inhibits the growth of mixed Blastocystis isolates.
We strongly recommend that the authors revise their claims throughout the manuscript (including the title, abstract, and discussion) to accurately reflect that the experiments were conducted on a co-culture, and that subtype-specific effects cannot be concluded from this study.
4.
We thank the authors for providing references regarding the use of eosin staining for protozoan viability. However, the response does not sufficiently address the core concerns regarding the methodology used for Blastocystis in this specific study.
Inappropriate Justification: The provided references demonstrate the use of 0.1% eosin staining for other amoebas like Giardia and Entamoeba histolytica. A method validated for one type of protozoan cannot be assumed to be equally effective and accurate for another, such as Blastocystis, without specific validation. The cellular characteristics and membrane permeability can differ significantly between species.
Lack of a Community Standard: The Trypan Blue exclusion assay is the widely recognized and more common method for assessing Blastocystis viability in the research community. The reliance on eosin staining, particularly if it is a method primarily established within the authors' previous studies, is not sufficient to be considered a standard without broader validation.
To ensure the credibility and reproducibility of the data, which is central to this study's conclusions, direct validation of the chosen method is required. We request that the authors perform a comparative experiment to validate their 0.1% eosin staining protocol against the conventional Trypan Blue exclusion assay for Blastocystis. The results of this comparison, demonstrating comparable accuracy and reliability, should be included in the Materials and Methods section. Without this direct validation, the accuracy of the viability data remains unsubstantiated, casting doubt on the reported inhibitory effects.
5.
We review the authors' addition to the discussion regarding the potential benefits of P. acidilactici-derived postbiotics. However, this literature-based discussion, while informative, does not address the critical question of the postbiotic's safety profile on host cells. A fundamental principle of therapeutic development is to demonstrate not only efficacy against a pathogen but also safety for the host.
The manuscript currently lacks experimental data on the cytotoxicity of the postbiotic towards human intestinal epithelial cells. It is entirely possible that the concentrations required to inhibit Blastocystis (e.g., the minimum concentration that inhibits 90%) could simultaneously induce damage to the gut epithelium. If the effective dose is also cytotoxic, the postbiotic would not be a viable therapeutic candidate.
Therefore, we strongly recommend that the authors perform a standard cytotoxicity assay (e.g., MTT or LDH assay) on a relevant human intestinal epithelial cell line (such as Caco-2 or HT-29). This assay should evaluate the effects of the postbiotic at the key concentrations used in the anti-protozoal experiments, including the MIC90 value. Without this essential safety data, any claims about the therapeutic potential of this postbiotic are premature.
Round 3
Reviewer 2 Report
Comments and Suggestions for Authors
1.
Upon reviewing the revised methods, we have identified a critical missing control group: uncultured medium control. The postbiotic solution was prepared by concentrating the broth after bacterial cultivation. However, the basal medium itself (MRS broth) contains complex components, such as organic acids and phenolic compounds, which, when concentrated, could inhibit the growth of Blastocystis.
Without a control group consisting of concentrated, uncultured MRS broth, it is impossible to definitively attribute the observed anti-protozoal activity to the bacterially-produced postbiotics rather than to the concentrated medium components themselves. This represents a major experimental confounder that could lead to erroneous conclusions. We request that the authors include this essential control group in their experiments to validate that the observed effects are indeed due to the postbiotic treatment.
2.
We acknowledge the authors' revision to describe their isolate as a "mixed Blastocystis isolate." However, this clarification raises a new and significant question regarding the stability of the co-culture. The authors state that PCR was used to identify subtypes ST1 and ST3 initially, with the postbiotic experiments conducted subsequently. Given the authors' own admission that their culture system is "not suitable for long-term subculturing," we are concerned that the subtypes may have lost during the subculturing period prior to the experiment.
To substantiate the claim that the treatment was tested on a mixed population, please clarify the number of subcultures between the initial identification and the final experiment. More importantly, please provide direct evidence (PCR methods) to confirm that both ST1 and ST3 were still present at the time the postbiotic treatment was initiated.
3.
“Technical and time constraints" are not a sufficient justification for omitting an experiment that is critical to supporting the author’s claims of therapeutic potential. Journals typically allow authors to request an extension for the revision period to complete critical experiments.
Furthermore, evaluating safety alongside efficacy is a standard and expected practice in Pathogens. For instance, a recent article in Pathogens (2024, DOI: 10.3390/pathogens13010080) assessing a natural extract included safety evaluations using an in vivo model. We insist that the authors conduct the in vitro cytotoxicity assay on a relevant epithelial cell line. Without this essential safety data, any claims about the therapeutic potential of this postbiotic remain incomplete. It is also not suitable with the authors' decision to complete the essential cytotoxicity assessment to future work.
